# Retinal Pigment Epithelial Cells Derived from Induced Pluripotent Stem (iPS) Cells Suppress or Activate T Cells via Costimulatory Signals

**DOI:** 10.3390/ijms21186507

**Published:** 2020-09-05

**Authors:** Sunao Sugita, Yoko Futatsugi, Masaaki Ishida, Ayaka Edo, Masayo Takahashi

**Affiliations:** 1Laboratory for Retinal Regeneration, RIKEN Center for Biosystems Dynamics Research, 2-2-3 Minatojima-minamimachi, Chuo-ku Kobe 650-0047, Japan; Yoko.futatsugi@riken.jp (Y.F.); masaaki.ishida@riken.jp (M.I.); ayaka.edo@riken.jp (A.E.); retinalab@ml.riken.jp (M.T.); 2Department of Ophthalmology, Kobe City Eye Hospital, 2-1-8 Minatojima-minamimachi, Chuo-ku, Kobe 650-0047, Japan; 3Department of Ophthalmology and Visual Science, Graduate School of Biomedical and Health Sciences, Hiroshima University, 1-2-3 Kasumi, Minami-ku, Hiroshima 734-8551, Japan

**Keywords:** iPS cells, retinal pigment epithelial cells, uveitis, suppression, stimulation, costimulatory molecules

## Abstract

Human retinal pigment epithelial (RPE) cells derived from induced pluripotent stem (iPS) cells have immunosuppressive properties. However, RPE cells are also known as immunogenic cells, and they have major histocompatibility complex expression and produce inflammatory proteins, and thus experience immune rejection after transplantation. In this study, to confirm the immunological properties of IPS-RPE cells, we examined whether human RPE cells derived from iPS cells could suppress or stimulate inflammatory T cells from uveitis patients via costimulatory signals. We established T cells from patients with active uveitis as target cells and used iPS-RPE cells as effector cells. As a result, cultured iPS-RPE cells inhibited cell proliferation and the production of IFN-γ by activated uveitis CD4^+^ T cells, especially Th1-type T cells. In contrast, iPS-RPE cells stimulated T cells of uveitis patients. The iPS-RPE cells constitutively expressed B7-H1/CD274 and B7-DC/CD273, and suppressed the activation of T cells via the PD-1 receptor. iPS-RPE expressed these negative costimulatory molecules, especially when RPE cells were pretreated with recombinant IFN-γ. In addition, iPS-RPE cells also expressed B7-H3/CD276 costimulatory molecules and activated uveitis T cells through the B7-H3-TLT-2 receptor. Thus, cultured iPS-derived retinal cells can suppress or activate inflammatory T cells in vitro through costimulatory interactions.

## 1. Introduction

Uveitis is a typical ocular inflammatory disease, and many uveitis patients are currently undergoing treatment. However, in cases of weak/poor inflammation in the eye, spontaneous remission is not uncommon. It has been speculated that the mechanism of spontaneous remission of inflammation is due to immune privilege in the eye [1,2,3]. In cases of weak inflammation, it is believed that intraocular infiltrating cells including T lymphocytes are suppressed, inactivated, or apoptotic, and are naturally excreted out of the eye. Retinal pigment epithelial (RPE) cells located in the posterior segment of the eye are also immune privileged cells and have a T-cell-suppressing effect. However, RPE cells, which are known to be immunogenic cells, have major histocompatibility complex (MHC) expression and experience rejection after RPE cell transplantation [4,5,6,7,8]. In other words, it is considered that RPE cells are highly likely to exhibit immunogenic (immunity promoting) or immunosuppressive effects, two properties of intraocular inflammation. For example, RPE cells express and produce a suppressive cytokine, transforming growth factor beta (TGFβ), to suppress T cells [9,10,11], while expressing and producing various inflammation-related factors. Representatives of the latter are complements [12,13,14], inflammatory chemokines such as CXCL9/10/11 [15], adhesion molecules such as CD54/ICAM1 [16], and MHC class I/II [17,18]. In addition, there are reports that RPE cells express costimulatory molecules [19,20], but the details of the mechanism behind this are still unknown.

Therefore, in this study, we examined whether human RPE cells derived from induced pluripotent stem (iPS) cells could suppress or stimulate uveitis T cells via costimulatory signals. iPS cell-derived RPE cells constitutively express negative costimulatory molecules such as B7-H1/CD274/programmed death-ligand 1 (PD-L1) or B7-DC/CD273/PD-L2, which can alter the expression of adaptive immune effector mechanisms in vitro. In addition, iPS-RPE cells also express positive costimulatory molecules such as B7-H3/CD276 to stimulate bystander T cells in vitro. For the in vitro assay, we established T cells from patients with active uveitis as target cells and used several RPE cell lines from human iPS cells as effector cells.

## 2. Results

### 2.1. Ability of IPS-Derived RPE Cells to Suppress or Activate T Cells from Active Uveitis Patients

We first tested whether iPS-RPE cells could suppress intraocular T cells (uveitis T cells) established from the peripheral blood mononuclear cells (PBMC) of patients with active uveitis. In the current study, we used four lines (836B1, 454E2, TLHD1, and 101G26) of human iPS-RPE cells for the in vitro assay. From FACS analysis, human iPS-RPE cells suppressed uveitis T cells (proliferation of CD4^+^ T cells) from the PBMC of patients with Vogt-Koyanagi-Harada (VKH) disease (Figure 1A). Similarly, iPS-RPE cells significantly suppressed IFN-γ production of Th1-type CD4^+^ uveitis T cells in patients with HLA-B27 associated acute anterior uveitis, acute retinal necrosis, CMV retinitis, sarcoidosis, VKH disease, and Behçet’s disease (Figure 1B). Although human iPS-RPE cells failed to suppress the T cells (rather than T-cell stimulation) when we prepared fewer RPE cells for the assay (i.e., T cell:RPE ratio = 200:1, 100:1 or 50:1), iPS-RPE cells significantly suppressed the IFN-γ production of T cells from the PBMC of active uveitis patients with VKH disease (Figure 1C). These results suggested that iPS-RPE cells have two immunological properties, immunosuppressive and immunogenic.

### 2.2. Survey of Candidate Immunosuppressive Molecules on IPS-RPE Cells

We hypothesized that costimulatory molecules expressed on the surfaces of RPE cells were involved in the activation or suppression of T cells. To detect molecules associated with costimulation, cell adhesion, and apoptosis, human iPS-RPE cells and primary RPE cells (fetal RPE) were subjected to flow cytometry after being stained with specific antibodies for the following candidate molecules: CD40, CD54 (ICAM1), CD70 (CD27 ligand), CD80 (B7-1), CD86 (B7-2), CD252 (OX40L), CD270 (HVEM), CD273 (B7-DC/PD-L2), CD274 (B7-H1/PD-L1), CD275 (ICOS-L/B7-H2), CD276 (B7-H3), B7-H4, Fas ligand (TNFSF6/CD95 ligand), 4-1BBL (TNFSF9/CD137 ligand), and GITRL (TNFSF18).

As a result, iPS-RPE cells and fetal RPE cells did not express CD40, CD70, CD80, CD86, OX40L, HVEM, ICOS-L, B7-H4, Fas-L, 4-1BBL, or GITRL. However, they constitutively expressed ICAM1, B7-H1, B7-H3, and small amounts of B7-DC (Figure 2). Also, the induction of ICAM1 and B7-H1 was higher in iPS derived RPE cells, as compared with the fetal RPE cells. These data suggested that the costimulatory molecule B7-H3 plays a role in the activation of T cells by its interaction with RPE cells. By contrast, B7-H1/B7-DC on RPE cells are negative costimulatory molecules, and ICAM1 on RPE cells is a cell adhesion molecule to bystander T cells [16].

### 2.3. Expression of B7-H1/CD274 and B7-DC/CD273 on Human IPS-RPE Cells

According to above results, the costimulatory molecules (B7-DC/CD273 and B7-H1/CD274) might play a role in the suppression/activation of T cell by iPS-RPE cells. Several groups previously reported that cultured human and mouse RPE cells constitutively express various costimulatory molecules, including B7-H1 [19,20]. In addition, these RPE cells inducibly express B7-H1 under inflammatory conditions, especially exposure to IFN-γ [19,20].

In FACS results, our established iPS-RPE cells, as well as primary fetal RPE cells, constitutively expressed B7-H1 costimulatory molecules, and the expression of this molecule was upregulated in iPS-RPE cells when these cells were cocultured with recombinant IFN-γ (Figure 3A). Similarly, human corneal endothelial cells and human fibroblasts also expressed B7-H1, but not iPS cells (Figure 3A).

In a dose-response assay with recombinant IFN-γ, we found that iPS-RPE cells expressed B7-H1 in a dose-dependent manner (Figure 3B). In addition, iPS-RPE cells exposed to supernatants collected from activated T cells from VKH uveitis patients highly expressed B7-H1 (Figure 3C). In IHC, both iPS-RPE without IFN-γ and IFN-γ-pretreated iPS-RPE cells clearly expressed B7-H1 on their surfaces (Figure 3D). From qRT-PCR analysis, we obtained similar results, i.e., iPS-RPE cells highly expressed B7-H1 mRNA compared with iPS cells (Figure 3E). Moreover, IFN-γ pretreatment of these iPS-RPE cells resulted in high expression of B7-H1 at the mRNA level compared with nontreated cells (Figure 3E).

To confirm these results, we used many human recombinant proteins, such as IL-1β, IL-1RA, IL-2, IL-4, IL-6, IL-8, IL-10, IL-12, IL-17A/F, IL-21, IL-22, IL-23, IL-27, TNF-α, TNFR1, IFN-γ, MIG, MCP-1, GM-SCF, TGFβ1, TGFβ2, TSP-1, MIF, and LPS. As shown in Appendix A, iPS-RPE cells exposed to IFN-γ, TNF-α, and IL-27 inducibly expressed B7-H1 according to FACS. IFN-γ, TNF-α, and IL-27 are all Th1-related cytokines [21].

We also examined whether iPS cell-derived RPE cells expressed B7-DC/CD273, since the cell surface receptor for B7-DC is PD-1, which is the same as for B7-H1 [22]. Although iPS-RPE cells poorly expressed B7-DC, IFN-γ-pretreated iPS-RPE cells clearly expressed it (Appendix A). These data suggest that iPS-RPE cells express costimulatory molecules (B7-H1 and B7-DC), especially under inflammatory conditions.

### 2.4. Expression of PD-1 on Uveitis T Cells

As a next step, we examined the expression of programed cell death 1 (PD-1: receptor for B7-H1/B7-DC) on target T cells. FACS analysis showed that naïve CD4 and CD8 T cells from PBMC of a healthy donor failed to express PD-1 receptor (Figure 4A). As revealed in Figure 4B, non-Th1-type CD4^+^ T cells from a VKH uveitis patient poorly expressed PD-1. However, Th1-type CD4^+^ T cells from uveitis patients with VKH disease or Behçet’s disease highly expressed PD-1 on their surfaces (Figure 4B). These results indicated that activated T cells, especially CD4^+^ Th1-type cells, greatly express the receptor for B7-H1/B7-DC costimulation.

### 2.5. Capacity of T-cell Suppression by Cocultures with Anti-PD-1 Antibody or PD-1 SiRNA

To determine whether B7-H1/B7-DC expressed by iPS-RPE cells could suppress bystander uveitis T cells, we examined cocultures with iPS-RPE cells and PBMC in the presence of antihuman PD-1 blocking antibody. As shown in Figure 5A, FACS analysis indicated that iPS-RPE cells suppressed the expression of IFN-γ in CD4^+^ T cells in PBMC from a VKH patient. In contrast, iPS-RPE cells failed to suppress the expression of IFN-γ in CD4^+^ T cells with the addition of anti-PD-1 blocking antibody in the cocultures (Figure 5A). We obtained similar results from IFN-γ ELISA using supernatants of T cells only, T cells plus iPS-RPE cells, and T cells plus iPS-RPE cells plus antibodies such as anti-PD-1 or isotype control (Figure 5B).

We next examined the effect of downregulating the mRNA expression of PD-1 on T cells by using siRNA. As revealed in Figure 5C, PD-1-siRNA-transfected uveitis T cells poorly expressed mRNA for PD-1, whereas control-siRNA-transfected T cells were able to clearly express the mRNA. Subsequently, we examined whether iPS-RPE cells were able to suppress the PD-1-siRNA-transfected T cells. The results indicated that iPS-RPE cells failed to suppress CD4^+^ T cells that were transfected with PD-1 siRNA (Figure 5D). Together, these results indicated that B7-H1/B7-DC on iPS-RPE cells is able to suppress bystander uveitis T cells through the PD-1 receptor.

### 2.6. Expression of B7-H3/CD276 on Human IPS-RPE Cells

As a next step, we examined the expression of B7-H3/CD276 on human iPS-RPE cells, because human iPS-RPE cells constitutively express this molecule (see Figure 2). Several reports [23,24,25] indicate that B7-H3 exhibits two properties, positive or negative costimulation of lymphocytes. First, we confirmed the expression of B7-H3 on several iPS-RPE cell lines by FACS and IHC analysis.

As shown in Figure 6A, human iPS-RPE cells, as well as primary fetal RPE cells, constitutively expressed this molecule, but the expression on IFN-γ-pretreated RPE cells (iPS-RPE and ES-RPE cells) was less compared with that of nontreated cells (Figure 6A), suggesting that this molecule might not affect inflammatory cytokines such as IFN-γ, unlike B7-H1 and B7-DC. According to IHC data, iPS-RPE cells (a 454E2 line) clearly expressed B7-H3 on their surfaces, but the expression of B7-H3 on fibroblasts from the same donor (454E2) was poor (Figure 6B). As well as control RPE cells (primary fetal RPE and ES-RPE cells), iPS-RPE cells greatly expressed the mRNA for B7-H3 compared with iPS cells according to qRT-PCR (Figure 6C). These results indicated that human RPE cells, including iPS-RPE cells, constitutively express B7-H3 costimulatory molecules.

### 2.7. Expression of TLT-2 on Uveitis T Cells

Although it is still controversial, the protein ‘triggering receptor expressed on myeloid cell-like transcript 2′ (TLT-2) on lymphocytes is considered a B7-H3 receptor [26,27,28]. In FACS analysis, T cells (CD4 and CD8), B cells (CD19), and monocytes/macrophages (CD11b) in PBMC from a healthy donor expressed TLT-2 receptor, especially on CD4^+^ T cells (Appendix A). Similar to PD-1 expression on T cells (see Figure 4), non-Th1-type CD4^+^ T cells from a VKH uveitis patient expressed TLT-2. However, Th1-type CD4^+^ T cells from the uveitis patient greatly expressed TLT-2 on their surfaces (Appendix A).

### 2.8. RPE Cells Derived from IPS Cells Can Activate Bystander T Cells via B7-H3 Costimulatory Signals

We previously demonstrated that monkey iPS-RPE cells express the B7-H3 costimulatory molecule, and these iPS-RPE cells were able to activate T lymphocytes via B7-H3 costimulatory signals [18]. To confirm this, we prepared recombinant B7-H3 proteins, B7-H3-siRNA-transfected iPS-RPE cells, and antihuman B7-H3 blocking antibodies. We first examined whether recombinant B7-H3 proteins could activate T cells in vitro. As expected, PBMC (Figure 7A) or CD4^+^ T cells (Figure 7B) produced significant amounts of IFN-γ in the presence of recombinant B7-H3 (2nd signal) and anti-CD3 (1st signal) antibodies. Moreover, CD8^+^ cytotoxic T cells produced significant amounts of granzyme B in the presence of recombinant B7-H3 and anti-CD3 antibodies (Figure 7C), suggesting that B7-H3 costimulatory signals can promote the activation of T cells.

We next examined whether B7-H3 costimulatory signals were necessary for iPS-RPE-recognition/stimulation by T cells. For the assay, antihuman B7-H3 antibodies were prepared for blocking. In the RPE-CD4 in vitro assay, CD4^+^ T cells produced less IFN-γ when iPS-RPE cells were cocultured with T cells plus anti-B7-H3 antibody, but the T cells produced large amounts of IFN-γ in the presence of anti-B7-H1 (PD-L1) antibody or isotype control (mouse IgG: Figure 7D). Moreover, as shown in Figure 7E, CD4^+^ T cells failed to produce IFN-γ when cocultured with RPE plus both anti-CD3 (1st signal blocking) and anti-B7-H3 (2nd signal blocking) antibodies. It is assumed that the major cellular interactions involved in T-cell activation by iPS-RPE cells are the MHC and B7-H3 pathways, because iPS-RPE cells express HLA (1st signal) and B7-H3 (2nd signal) costimulatory molecules.

As a final step, we examined whether downregulation of B7-H3 on iPS-RPE cells could stimulate uveitis T cells in vitro. Compared with control-siRNA-transfected iPS-RPE cells, the expression of B7-H3 mRNA in B7-H3-siRNA-transfected RPE cells was strongly downregulated according to FACS (Figure 8A) and qRT-PCR (Figure 8B). In in vitro assays, B7-H3-siRNA-transfected iPS-RPE cells failed to stimulate uveitis T cells, while control-siRNA RPE cells actually activated uveitis T cells in vitro (Figure 8C). Taken together, we summarize the relationship between RPE cells and T cells with 2nd costimulatory signals in Appendix A. B7-H3 costimulation on RPE cells can activate T cells through receptors such as TLT-2. In contrast, B7-H1/B7-DC costimulatory molecules on RPE cells can suppress T cells through the PD-1 receptor. IFN-γ inflammatory cytokines secreted by T cells are necessary for the latter interaction. In fact, uveitis T cells are able to secrete many inflammatory cytokines [29,30].

## 3. Discussion

In the present study, we showed that cultured human iPSC-derived RPE cells significantly inhibited T cell activation, e.g., proliferation and cytokine production. iPS-RPE cells significantly suppressed the activation of intraocular T cells from patients with active uveitis. The iPS-RPE cells constitutively expressed negative costimulatory molecules such as B7-H1. iPS-RPE cells inducibly expressed B7-H1 and B7-DC under IFN-γ exposure. These findings demonstrate that iPS-RPE cells can actively suppress the proliferation of Th1-type CD4^+^ T cells that produce IFN-γ from patients with ocular inflammation and suggest that iPS-RPE cells modify T cell function by modulating the production of the effector cytokine IFN-γ. In fact, the aqueous humor of patients with active ocular inflammation contains high levels of Th1 effector cytokines [30]. Interestingly, iPS-RPE cells could also stimulate CD4^+^ T cells from uveitis patients. iPS-RPE positively expressed the costimulatory molecule B7-H3. In addition, iPS-RPE cells activated uveitis CD4^+^ T cells through the B7-H3-TLT-2 receptor. Thus, cultured iPS-derived retinal cells can suppress or activate uveitic T cells in vitro through costimulatory interactions.

Molecular homologs of B7-like ligands such as B7-H1 (CD274/PD-L1) and B7-DC (CD273/PD-L2) have been identified [22,31,32]. These costimulatory molecules have been shown to downregulate T cell activation through PD-1 with negative costimulatory signals [33]. PD-1 is expressed by CD4^+^ T cells (especially activated Th1 cells), CD8^+^ T cells, B cells, and monocytes [33]. The expression of B7-H1 is promoted by the inflammatory cytokine IFN-γ [31,32]. Therefore, B7-H1 may interact with PD-1-positive cells to achieve the suppression of inflammation, including T-cell immunity.

In a previous report [34], human corneal endothelial cells efficiently suppressed the proliferation of Th1 cells that expressed PD-1 among activated T cells established from patients with uveitis or corneal endotheliitis. In the current study, we obtained similar results. For example, we showed that neutralizing antibodies for PD-1 blocked the suppressive effect of iPS-RPE cells on Th1 cells. In addition, iPS-RPE cells failed to suppress PD-1-siRNA-transfected T cells from CD4-positive uveitis T cells in vitro. These results indicated that iPS-RPE cells are able to impair the effector functions and activation of Th1-type CD4^+^ T cells via the B7-H1/B7-DC-PD-1 interaction.

Pigment epithelial cells in the eye have immunosuppressive properties. Murine iris pigment epithelial cells express B7-2 to suppress bystander T cells [35]. In addition, regulatory T cells induced by iris pigment epithelial cells express B7-2 and suppress T cells via the B7-2-CTLA-4 interaction [36,37]. In a cornea study, Hori et al. reported that B7-H1 expressed on corneal endothelial cells leads to the suppression of corneal allograft rejection in a mouse model [38]. Pigment epithelial cells in the posterior segment in the eye exhibit similar immunoregulatory properties against activated T lymphocytes. Murine and human RPE cells express B7-H1 in order to suppress bystander T cells via PD-1 [19,20]. Our results and other investigations support the hypothesis that RPE cells and corneal cells may contribute to the maintenance of the privileged immune status of the eye by inducing peripheral immune tolerance.

Although there was no expression of B7-1 and B7-2 costimulatory molecules on iPS-RPE cells (Figure 2), the cells constitutively expressed MHC and the costimulatory molecule B7-H3 (Figure 2 & Figure 6), and they can activate T cells through the 1st and 2nd signals. T cells were not activated if the RPE cells were cultured with anti-B7-H3 blocking antibody or B7-H3 siRNA (downregulating B7-H3 expression). Although the function of B7-H3 is still controversial, it can promote T-cell-mediated immune responses and the development of acute and chronic immune rejection in allografts after transplantation [39]. Our results suggest that costimulatory molecules on RPE cells mediated allogeneic T-cell activation. As a strategy for allografts of RPE transplantation (HLA mismatched) in human trials, the administration of B7-H3 blockade might be useful, because T cells do not respond to RPE cells in the presence of anti-B7-H3 blocking antibody. In fact, we recently reported that one retinal patient had immune rejection of HLA-matched iPS-RPE cell transplantation [8]

TLT-2 (TREML2) is a type I transmembrane protein in the TREM family. TLT-2 is expressed on B cells, myeloid cells, and macrophages, and it has also been found to be expressed on CD8^+^ T cells and activated CD4^+^ T cells [26,27,28]. TLT-2 can costimulate the activation of T cells. Transduction of TLT-2 into T cells results in enhanced IL-2 and IFN-γ production (maybe in Th1 cells). It has been reported that B3-H7 is a receptor of TLT-2 [26,27,28], and as revealed in the current study, uveitis T cells, especially Th1 cells, highly express TLT-2 (Appendix A). Thus, it is possible that B7-H3 blockade may prevent severe intraocular inflammation such as in autoimmune uveitis. 

In recent years, cancer treatment targeting PD-1 (nivolumab or pembrolizumab) has been performed, but there are reports of autoimmune uveitis as a side effect [40,41,42,43,44]. Obata et al. reported VKH disease-like autoimmune pan-uveitis associated with serous retinal detachment [42], and anti-PD-1 treatment was discontinued because of headaches in the patient. The retinal detachment disappeared within 3 months after starting topical corticosteroid treatment. Matsuo et al. reported similar uveitis [44]. Thus, anti-PD-1 antibody treatment causes autoimmune intraocular inflammatory disease. This means that PD-1 enhancers can be used to suppress uveitis T cells, because B7-H1, which is a ligand for PD-1, is constitutively expressed in ocular tissues such as retina/RPE [19,20] and cornea [34,38]. Alternatively, anti-B7-H3 or anti-TLT-2 antibody therapy may be used to treat uveitis. In any case, antibody therapy and biologics can be useful in the treatment of uveitis if used properly.

## 4. Material and Methods

### 4.1. Establishment of IPS-Derived RPE (iPS-RPE) Cells

We established four lines of human iPS cells, i.e., 836B1, 101G26, 454E2, and TLHD1, as previously reported [11,17]. For differentiation into RPE cells, human iPS cells were cultured on gelatin-coated dishes in GMEM medium with several supplements including KnockOut Serum Replacement (KSR; Life technologies, Tokyo, Japan) [11,17]. Signal inhibitors Y-27632 (Wako, Osaka, Japan), SB43542 (Sigma, St. Louis, MO, USA), and CKI-7 (Sigma) were added to the GMEM. After the appearance of RPE-like colonies, the medium was changed to Dulbecco’s modified Eagle’s medium (DMEM; Wako) with B27 supplement (Invitrogen, Carlsbad, CA, USA) and L-glutamine (Sigma), and the colonies were transferred to CELLstart^TM^ (Invitrogen)-coated dishes in B27 with basic fibroblast growth factor (FGF; Wako) and SB431542 [11,17]. Our established iPS-RPE cell lines were hexagonal morphology and pigment containing cells. Besides, they expressed gap junction proteins (ZO-1) in immunohistochemistry, and expressed RPE-specific markers such as RPE65, pigment epithelium derived factor (PEDF), tyrosinase, and so on in PCR.

The controls used for iPS-RPE cells were human ES-derived RPE cells [45] and human primary fetal RPE cells (Ronza, Tokyo, Japan).

This research study followed the tenets of the Declaration of Helsinki, and the study was approved by the Institutional Ethics Committee of the Kobe City Eye Hospital/Kobe City Medical Center General Hospital (Approval No. ezn190202, 23 January 2019).

### 4.2. Establishment of T Cells from Uveitis Patients

T cell lines of patients with active uveitis such as HLA-B27 associated acute anterior uveitis, acute retinal necrosis, cytomegalovirus (CMV) retinitis, sarcoidosis, Vogt-Koyanagi-Harada (VKH) disease, and Behçet’s disease were established from peripheral blood mononuclear cells (PBMC) by a previous method [46]. We established Th1 cell lines from patients with VKH disease using antihuman CD3 agonistic antibody, recombinant IL-12, recombinant IFN-γ, and antihuman IL-4 antibody.

They expressed T-bet^+^IFN-γ^+^Stat1^+^CXCR3^+^ according to qRT-PCR and secreted large amounts of IFN-γ according to ELISA, and they were CD4^+^IFN-γ^+^ cells according to FACS analysis.

As additional targets, activated T cells were established from T cells of PBMC from healthy donors. CD4-positive or CD8-positive T cells were prepared separately by using separation beads and columns (MACS cell isolation kit; Miltenyi Biotec, Auburn, CA). More than 96% of these cells were CD4/CD8-positive. T cells were cocultured in RPMI1640 medium containing 10% fetal bovine serum (FBS) plus recombinant IL-2 (100 U/mL; BD Biosciences, San Jose, CA, USA). Purified T cells from healthy donor PBMC were also prepared for assays. Uveitis T cells (or T cells from healthy donors) were stimulated with antihuman CD3 agonistic antibody (1 μg/mL: Clone UCHT1; Ancell, Bayport, MN) and were then incubated with iPS-RPE cells for 48 or 72 h. iPS-RPE cells were cultured separately in 96-well plates (5 × 10^5^ cells/well) or 24-well plates (2 × 10^6^ cells/well). The supernatants of cocultures with iPS-RPE cells and uveitis T cells were measured using human IFN-γ ELISA (R&D Systems, Minneapolis, MN).

For the blocking in some in vitro experiments, antihuman B7-H1 or antihuman B7-H3 antibodies were used as neutralizing antibodies in cocultures with iPS-RPE cells and T cells. The information for all antibodies is provided in Appendix A.

### 4.3. Flow Cytometry

T-cell activation was assessed in terms of proliferation by an antihuman Ki-67 antibody (BioLegend, San Diego, CA). After 72 h, Ki-67-labeled T cells were washed and analyzed by flow cytometry. Antihuman IFN-γ antibody (R&D System) was also used for staining and examination by FACS analysis. The antibodies of immune cells such as CD4, CD8, CD11b, and CD19 were also used in FACS analysis.

The expression of costimulatory molecules on human iPS-RPE cells and control cells (primary RPE cells) was examined by FACS analysis for CD40, CD54 (ICAM-1), CD70 (CD27 ligand), CD80 (B7-1), CD86 (B7-2), CD252 (OX40L), CD270 (HVEM), CD273 (PD-L2/B7-DC), CD274 (PD-L1/B7-H1), CD275 (ICOS-L/B7-H2), CD276 (B7-H3), B7-H4, Fas ligand (CD95 ligand/TNFSF6), 4-1BBL (CD137 ligand/TNFSF9), and GITRL (TNFSF18).

The expression of costimulatory receptors such as programmed cell death-1 (PD-1) or triggering receptor expressed on myeloid cell-like transcript 2 (TLT-2, TREML2) on target uveitis cells or control cells from healthy donors was also examined by FACS analysis. We previously reported the procedure for RPE cell and T cell (PBMC) staining in FACS analysis [11]. The information for all antibodies is provided in Appendix A. Some results are expressed as the mean fluorescence intensity (MFI). Samples were analyzed by a FACSCanto^TM^ II or FACSAria^TM^ II flow cytometer (BD Biosciences).

### 4.4. Quantitative RT-PCR

Expression of mRNA for B7-H1/CD274, B7-DC/CD273, or B7-H3/CD276 in iPS-RPE cells was evaluated using quantitative RT-PCR (qRT-PCR). In qRT-PCR for B7-H1 or B7-DC, total RNA was isolated from human iPS-RPE cell lines (*n* = 3) or control human iPS cells (*n* = 1). Total RNA was also isolated from human iPS-RPE cells that were pretreated with recombinant IFN-γ (100 ng/mL). In qRT-PCR for B7-H3, total RNA was isolated from human iPS-RPE cells (*n* = 4), control RPE cells (*n* = 2), or control human iPS cells (836B1). After cDNA synthesis, the expressions of B7-H1, B7-DC, and B7-H3 and β-actin in triplicate samples were analyzed by qRT-PCR with a LightCycler 480 instrument using qPCR Mastermix and Universal Probe Library assays (all Roche Diagnostics, Mannheim, Germany) as previously reported [11]. Relative mRNA expression was calculated with Relative Quantification Software in Roche Diagnostics by using an efficiency-corrected algorithm with standard curves and reference gene normalization against β-actin (delta delta cycle threshold (Ct): ΔΔCt), and results indicate the relative expression of the molecules (ΔΔCt: control cells = 1) [11]. The information for all primers and probes is provided in Appendix A.

### 4.5. Transfection of SiRNA

SiRNA targeting human B7-H3/CD276 or control-siRNA (both Santa Cruz, CA) was transfected into human iPS-RPE cells [11]. On day 0, the cells were cultured in DMEM containing antibiotic free 5% FBS. After overnight culture, iPS-RPE cells were transfected with B7-H3-siRNA reagent at 37 °C for 6 h and then cultured in DMEM + 10% FBS at 37 °C for 24 h. After incubation, the cells were harvested and examined for the expression of B7-H3 mRNA by qRT-PCR or FACS analysis. We also prepared PD-1-transfected CD4^+^ T cells from PBMC of uveitis patients using similar methods (human PD-1-siRNA; Santa Cruz). The expression of PD-1 on CD4^+^ T cells was assessed by qRT-PCR.

### 4.6. Immunohistochemistry

The expression of B7-H1/CD274 or B7-H3/CD276 in iPS-RPE cells was evaluated by immunohistochemistry (IHC) [11]. In IHC for B7-H1, iPS-derived RPE cells and IFN-γ-treated iPS-RPE cells were fixed with 4% PFA-PBS for 15 min at room temperature, washed 3 times with PBS, and permeabilized with 0.3% Triton X-100-PBS. In IHC for B7-H3, iPS-derived RPE cells and control human fibroblasts were also fixed with a similar procedure. Antihuman B7-H1 antibody (eBioscience), antihuman B7-H3 antibody (R&D Systems), rat IgG, and mouse IgG were used as the primary antibodies. Then, anti-mouse IgG or antirat IgG were used as the secondary antibodies (Alexa Fluor 488 or 546; Invitrogen). Cell nuclei were counterstained with 4′,6-diamidino-2-phenylindole (DAPI). After washing, stained cells were observed by fluorescence microscopy [11]. These antibodies are listed in Appendix A.

### 4.7. Recombinant Proteins

Recombinant human IFN-γ proteins (R&D Systems) were used for RPE cell experiments, and recombinant human proteins were used for the iPS-RPE cells: IL-2, IL-1β, IL-12, IL-1RA, IL-4, IL-6, IL-8/CXCL, IL-10, IL-17A/F, IL-21, IL-22, IL-23, IL-27, TNF-α, soluble TNFR1, MIG/CXCL9, MCP-1/CCL2, GM-CSF, TGFβ1, TGFβ2, thrombospondin-1 (TSP-1), macrophage migration inhibitory factor (MIF), and lipopolysaccharide (LPS). In addition, recombinant human B7-H3/CD276 proteins were used for T cell experiments in vitro. The information for all recombinant proteins is provided in Appendix A.

### 4.8. Statistical Evaluation

All experiments were repeated at least twice, and the data are presented as means ± SEM. Student’s paired or unpaired *t* test was used for all statistical analyses (GraphPad Software). Values were considered statistically significant if *p* was less than 0.05.

## 5. Conclusions

Human iPS-RPE cells expressing the costimulatory molecule B7-H1/B7-DC selectively suppressed Th1 cells that showed high expression of PD-1. By contrast, these RPE cells expressing another costimulatory molecule, B7-H3, activated T cells that also expressed TLT-2. Ocular infiltrating cells in the retina appear in the subretinal space when a patient has uveitis. Effector T cells are converted into inactivated cells or activated cells during their migration through the subretinal space. When exposed to IFN-γ inflammatory cytokines, RPE cells show marked induction of B7-H1/B7-DC expression, and T cells exposed to RPE cells can be inactivated since intraocular T cells express high levels of PD-1. Conversely, the response of infiltrating T cells to the influence of retinal cells promotes T cell activation via B7-H3 costimulation. The fate of ocular infiltrating cells in the retina will depend on what factors the intraocular cells express and what signals they are given. It seems that the intraocular environment of the individual is greatly involved.

## Figures and Tables

**Figure 1 ijms-21-06507-f001:**
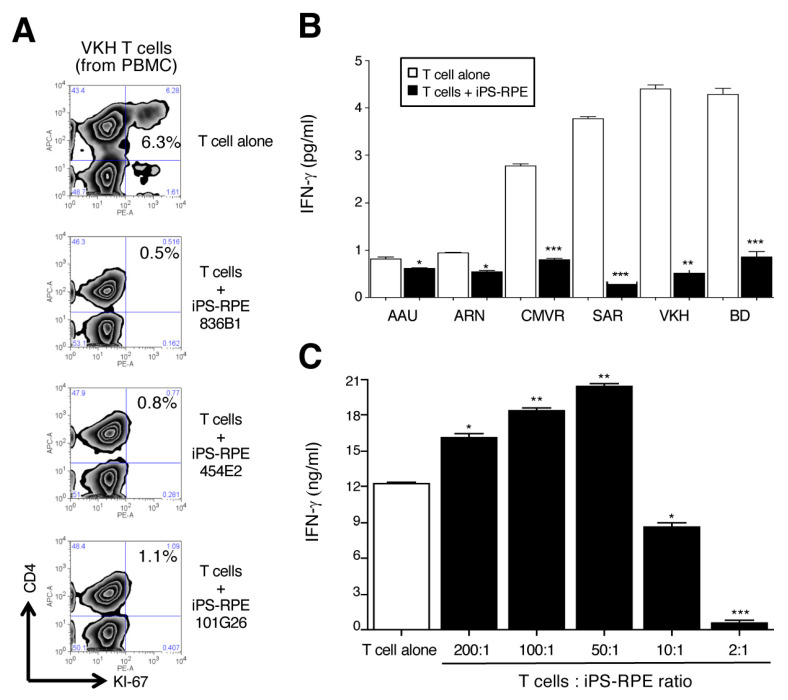
Capacity of iPS-derived RPE to suppress or stimulate uveitis T cells in vitro. (**A**) Three lines of established human iPS-RPE cells were used (836B1, 454E2, and 101G26) for the in vitro assays. Uveitis T cells were established from PBMC of a VKH disease patient with active uveitis. The proliferation of CD4^+^ T cells in VKH PBMC in the presence of iPS-RPE cells was evaluated by FACS analysis (CD4 and Ki-67 staining). PBMC were cocultured with to 5 × 10^5^ (T cell:RPE ratio = 2:1) iPS-RPE cells. Numbers in the histograms indicate the percentage of cells double-positive for CD4 and Ki-67. (**B**) Data of IFN-γ-production in supernatants from Th1-type CD4^+^ uveitis T cells + iPS-RPE cells. T cells were cocultured with to 5 × 10^5^ (T cell:RPE ratio = 2:1) iPS-RPE cells. Open bars indicate T cells alone, and black bars indicate T cells plus iPS-RPE cells. AAU: HLA-B27 associated acute anterior uveitis; ARN: acute retinal necrosis; CMVR: CMV retinitis; SAR: sarcoidosis; VKH: Vogt-Koyanagi-Harada disease; BD - Behçet’s disease. Data are the means ± SEM of three ELISA determinations. (**C**) Capacity of iPS-RPE cells to suppress the activation of bystander T cells in a T-cell number assay. In the presence of antihuman CD3 agonistic antibody and rIL-2, purified CD4^+^ T cells from PBMC of active uveitis patients with VKH disease were cocultured with 5 × 10^3^ to 5 × 10^5^ (T cell:RPE ratio = 200:1, 100:1, 50:1, 10:1, or 2:1) iPS-RPE cells (454E2) for 48 h. Results indicate the IFN-γ production by T cells exposed to iPS-RPE cells. Data are the means SEM of 3 ELISA determinations. * *p* < 0.05, ** *p* < 0.005, *** *p* < 0.0005, as compared to the positive control (T cells alone: open bar).

**Figure 2 ijms-21-06507-f002:**
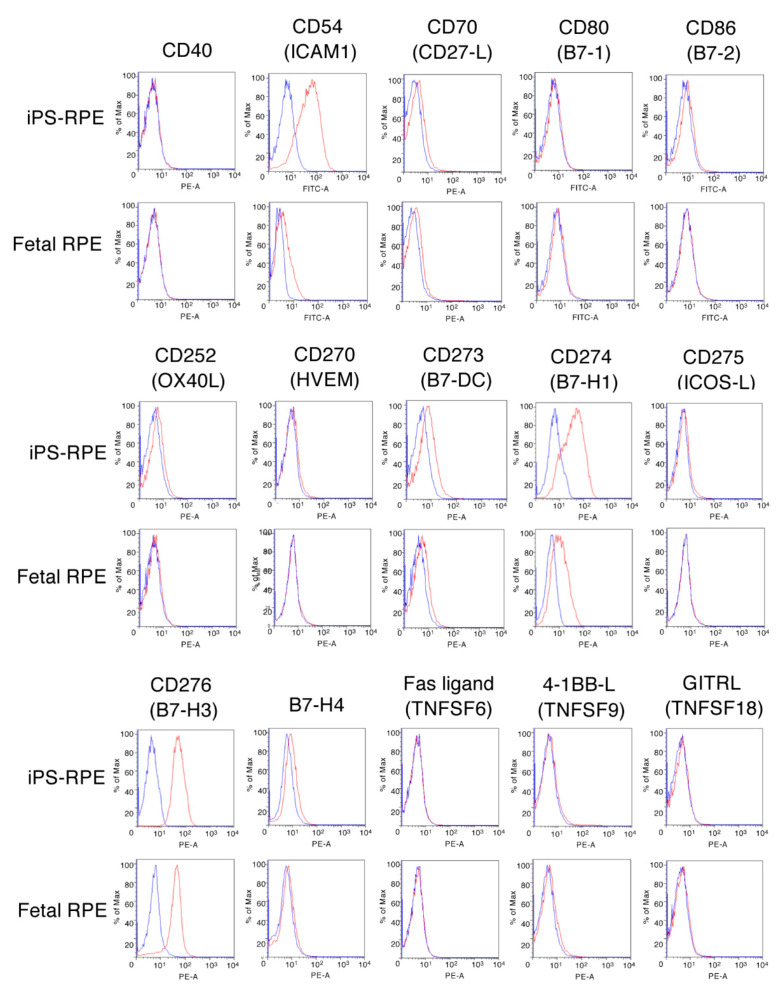
Expression of costimulatory molecules on human iPS cell-derived RPE cells. Human iPS-RPE cells (836B1) or control primary human RPE cells (fetal RPE cells) were stained with antihuman CD40, CD54 (ICAM1), CD70 (CD27 ligand), CD80 (B7-1), CD86 (B7-2), CD252 (OX40L), CD270 (HVEM), CD273 (PD-L2/B7-DC), CD274 (PD-L1/B7-H1), CD275 (ICOS-L/B7-H2), CD276 (B7-H3), B7-H4, Fas ligand (CD95 ligand/TNFSF6), 4-1BBL (CD137 ligand/TNFSF9), and GITRL (TNFSF18) antibodies. We obtained similar results with other iPS-RPE cells (454E2, TLHD1). Blue histogram: data for isotype control. PE-A—Phycoerythrin A.

**Figure 3 ijms-21-06507-f003:**
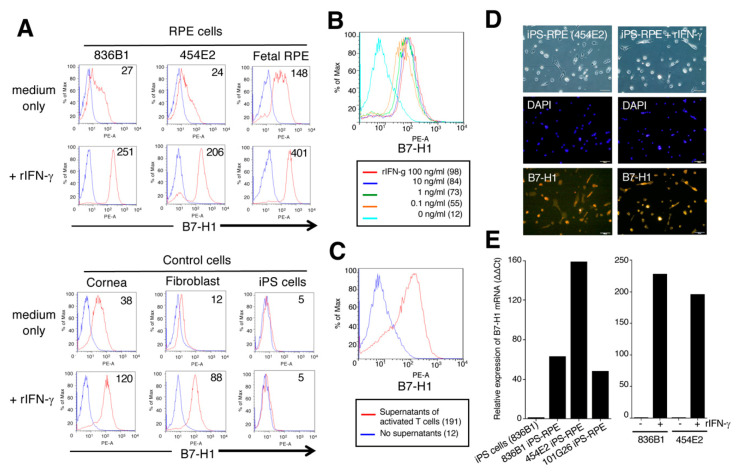
Expression of B7-H1 molecules on human iPS-RPE cells. The expression of B7-H1 costimulatory molecules on established iPS-RPE cells was assessed by FACS (**A**–**C**), IHC (**D**), and qRT-PCR (**E**). (**A**) iPS-RPE cells, 836B1 and 454E2, and primary fetal RPE cells as a control; lower histograms are cells with recombinant IFN-γ. In FACS analysis, human corneal endothelial cells (Cornea) and fibroblast and control human iPS cells (454E2 line) were also prepared. Blue histogram: data for isotype control. The number in the histogram indicates mean fluorescence intensity (MFI). (**B**) Dose-response assays with recombinant IFN-γ used 0, 0.1, 1.0, 10, and 100 ng/mL. The numbers in parentheses indicate MFI. (**C**) We also prepared iPS-RPE cells exposed to supernatants collected from activated T cells from a VKH uveitis patient (red histogram). Blue histogram: no supernatants (control). The numbers in parentheses indicates MFI. (**D**) For IHC, we also prepared 454E2 iPS-RPE cells with or without recombinant IFN-γ. The expression of B7-H1 is shown in red, and middle pictures show cell nuclei counterstained with DAPI. Scale bars, 100 μm. (**E**) The expression level of B7-H1 mRNA in iPS-RPE cells, 836B1, 454E2, 101G26, and control iPS cells (836B1) is shown. The right panel indicates iPS-RPE cells (836B1, 454E2) with or without recombinant IFN-γ. Results indicate the relative expression of B7-H1 (ΔΔCt: control iPS cells = 1).

**Figure 4 ijms-21-06507-f004:**
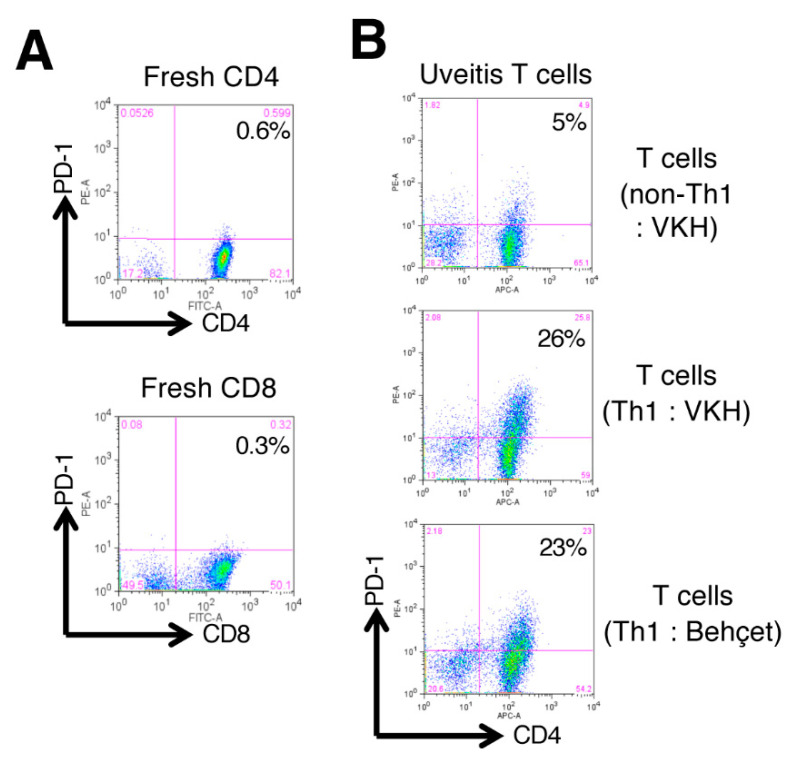
Expression of PD-1 on T cells from uveitis patients. (**A**) Expression of PD-1 (receptor for B7-H1/B7-DC) on T cells from PBMC of a healthy donor. Upper histogram: fresh naïve CD4^+^ T cells; lower histogram: fresh naïve CD8^+^ T cells by FACS analysis. (**B**) Expression of PD-1 on non-Th1-type CD4^+^ T cells from a VKH uveitis patient, and Th1-type CD4^+^ T cells from uveitis patients with VKH disease or Behçet’s disease. Numbers in the histograms indicate the percentage of cells double-positive for CD4/CD8 and PD-1.

**Figure 5 ijms-21-06507-f005:**
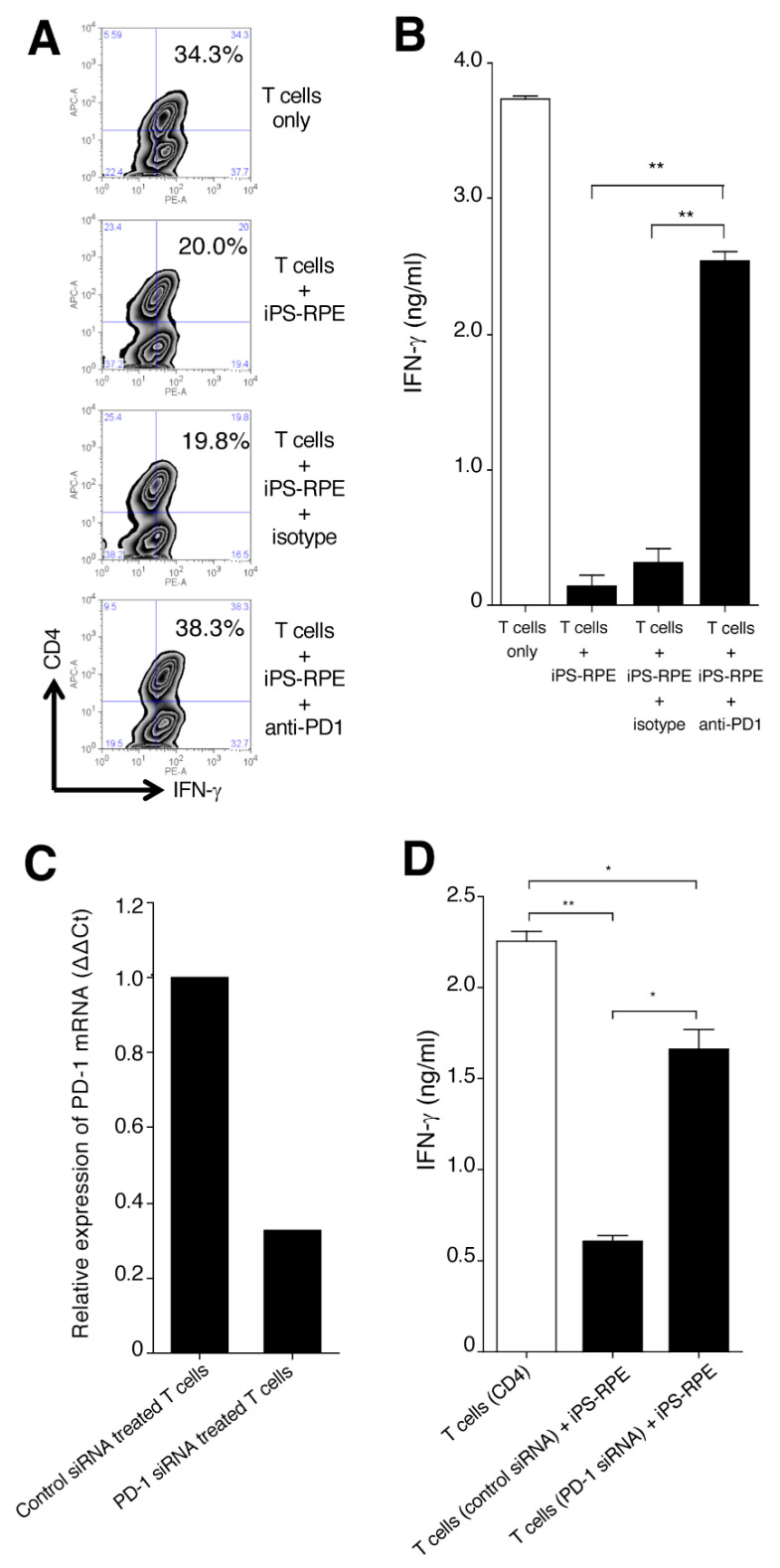
Effect of B7-H1 produced by human iPS-RPE cells on the suppression of T-cell activation. CD4^+^ Th1 cells from a VKH uveitis patient were cocultured with iPS-RPE cells in the presence of anti-PD-1 blocking antibody or isotype control and evaluated by IFN-γ staining (**A**: FACS) or IFN-γ production (**B**: ELISA) by the T cells. T cells were cocultured with to 5 × 10^5^ (T cell:RPE ratio = 2:1) iPS-RPE cells. (**A**) In FACS, numbers in the histograms indicate the percentage of cells double-positive for CD4 and IFN-γ. (**B**) Data are the means ± SEM of 3 ELISA determinations. ** *p* < 0.005, as compared to two groups. (**C**) PD-1-siRNA-transfected CD4^+^ Th1 cells from a VKH patient were harvested on day 3 and examined for the expression of PD-1 mRNA by qRT-PCR. Control-siRNA-transfected T cells were also analyzed. Results indicate relative expression (ΔΔCt). (**D**) PD-1-siRNA-transfected CD4^+^ T cells (or control-siRNA-transfected cells) were cocultured with iPS-RPE cells and evaluated by IFN-γ production by the uveitis T cells (T cell:RPE ratio = 2:1). Data are the means ± SEM of 3 ELISA determinations. * *p* < 0.05, ** *p* < 0.005, as compared to 2 groups. These data are representative of 3 individual experiments.

**Figure 6 ijms-21-06507-f006:**
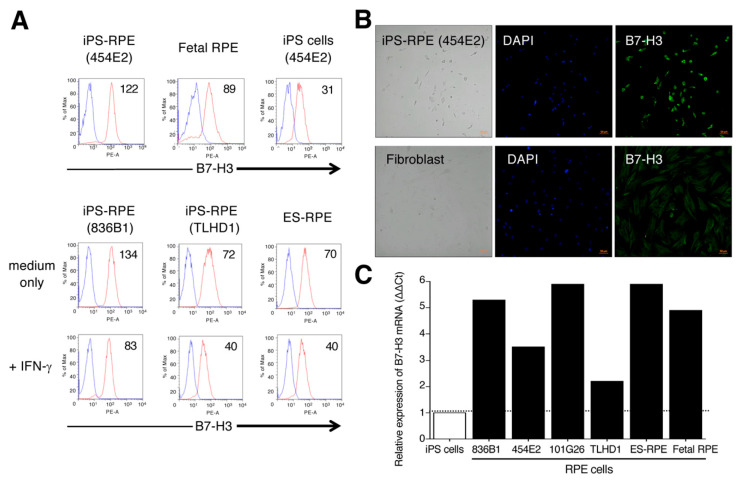
Detection of B7-H3 costimulatory molecules on human iPS-RPE cells. (**A**) The expression of B7-H3 on iPS-RPE cells was assessed by FACS analysis. 454E2 iPS-RPE cells, fetal RPE cells, and 454E2 iPS cells as a control were used. In the analysis, IFN-γ-pretreated RPE cells (iPS-RPE cells or ES-RPE cells) were also prepared. The numbers in the histograms indicate MFI. Blue histogram: data for isotype control. (**B**) Detection of B7-H3 on iPS-RPE cells by immunostaining. iPS-RPE cells (454E2) clearly expressed B7-H3 but fibroblasts did not. Cell nuclei were counterstained with DAPI. Scale bars, 50 μm. (**C**) In qRT-PCR for B7-H3 mRNA, PS-RPE cells, 836B1, 454E2, 101G26, TLHD1, ES-RPE cells, fetal RPE cells, and control iPS cells were prepared. Results indicate the relative expression of B7-H3 (ΔΔCt: control iPS cells = 1).

**Figure 7 ijms-21-06507-f007:**
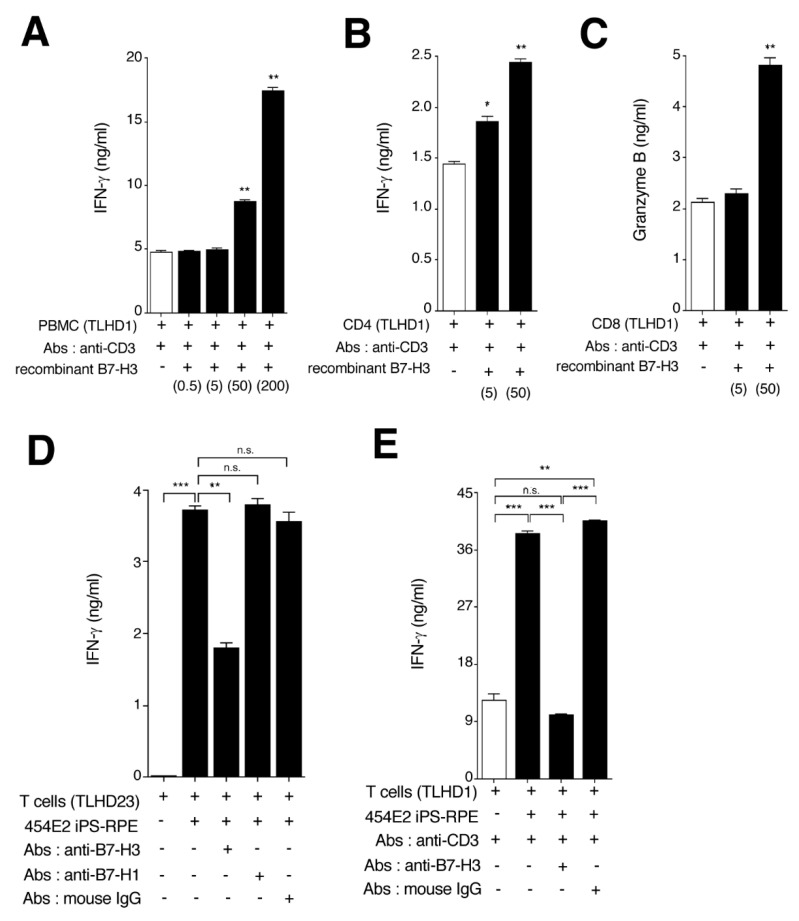
Capacity of human iPS-RPE cells to stimulate the activation of bystander T cells via B7-H3. Recombinant human B7-H3 proteins were used in in vitro assays with lymphocytes. (**A**) PBMC were cocultured with recombinant B7-H3 (0.5. 5, 50, or 200 ng/mL) in the presence of antihuman CD3 antibody for 72 h. The supernatants were harvested for IFN-γ ELISA. (**B**) Purified CD4^+^ T cells were cocultured with recombinant B7-H3 (5 or 50 ng/mL) and evaluated by IFN-γ ELISA. (**C**) Purified CD8^+^ T cells were cocultured with recombinant B7-H3 (5 or 50 ng/mL) and evaluated by granzyme B ELISA. Data are the means ± SEM of 3 ELISA determinations. * *p* < 0.05, ** *p* < 0.005, as compared to positive controls (open bar: no recombinant B7-H3). Data are representative of 3 individual experiments. (**D**) CD4^+^ T cells were cocultured with iPS-RPE cells (454E2) in the presence of antibodies such as anti-B7-H3, anti-B7-H1, or isotype control (mouse IgG) and assessed by IFN-γ ELISA (T cell:RPE ratio = 100:1). (**E**) CD4^+^ T cells were also cocultured with iPS-RPE cells in the presence of anti-CD3 (1st signal blocking) plus anti-B7-H3 (2nd signal blocking) and assessed by IFN-γ ELISA (T cell:RPE ratio = 100:1). Data are the means ± SEM of 3 ELISA determinations. ** *p* < 0.005, *** *p* < 0.0005, as compared to 2 groups. n.s., not significant.

**Figure 8 ijms-21-06507-f008:**
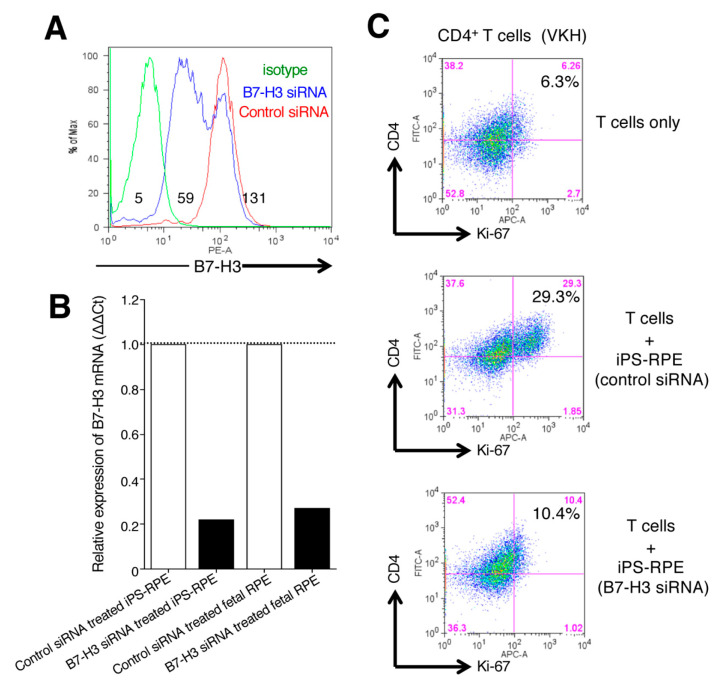
Effect of B7-H3 expression by iPS-RPE cells on the stimulation of T-cell activation. (**A**) B7-H3-siRNA-transfected iPS-RPE cells (TLHD1) were harvested on day 3 and examined for expression of B7-H3 by flow cytometry. Control-siRNA-transfected cells were also analyzed. The numbers in the histograms indicate MFI. Data are representative of 2 experiments. (**B**) B7-H3-siRNA-transfected iPS-RPE cells (TLHD1) or control RPE cells (fetal RPE cells) were examined for expression of B7-H3 mRNA by qRT-PCR. Data are representative of 3 experiments. Control-siRNA-transfected cells were also analyzed (ΔΔCt: control cells = 1). (**C**) CD4^+^ T cells from a VKH disease patient were cocultured with B7-H3-siRNA-transfected iPS-RPE (or control-siRNA-transfected) cells and evaluated by Ki-67 staining (for proliferation) of T cells. T cells were cocultured with to 2 × 10^4^ (T cell:RPE ratio = 100:1) iPS-RPE cells. Data are representative of 3 individual experiments. Numbers in the histograms indicate the percentage of cells double-positive for CD4 and Ki-67.

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
