# Peer review of "Retinal Pigment Epithelial Cells Derived from Induced Pluripotent Stem (iPS) Cells Suppress or Activate T Cells via Costimulatory Signals"

_ijms, 2020, doi:10.3390/ijms21186507_

Round 1

Reviewer 1 Report

The authors describe the immunomodulatory properties of iPS-derived RPE cells and its implications on interaction with immune cells from uveitis patients. The following points need attention.

  1. Line 366. 836B1, 101G26, 454E2 and TLHD1 are in fact the RPE cells derived from iPS cells, but their description on line 366 appears as if these were only the human iPS cells, and not the ones that were already converted into RPE. Related to this, it would be nice if the authors describe the details on the appearance of these cells after they become RPE like. Cuboidal morphology, distribution of gap junction proteins, or a description of similar markers will strengthen the conversion into RPE like cells.
  2. Figure 2. The authors claim only ICAM1 and B7-H1 as the markers that were induced. They claim B7-H3 to be a weakly induced, while in fact, it is expressed at the same level as high expressers. Can the authors clarify this? Also, the induction of ICAM1 and B7-H1 is higher in iPS derived cells, as compared with the fetal RPE cells. This deserves a mention.
  3. Line 121. The title of this paragraph includes only B7-H1, while the following description also has B7-DC and B7-H3 included. Why was the title restricted to only B7-H1?
  4. The relative expression of target genes is described in the following figures (Fig 3E, Fig 5C, Fig 6C, Fig 8B, Fig S2B and Fig S2C). The authors have given Expression of (target) mRNA and in parenthesis, they have delta delta C as the title of y-axis, and used the same description in figure legends. Again, what is described here is relative expression of the specific genes, with a control arbitrarily taken as 1. In the calculation, delta delta C does not in fact give the number that is provided in the figure, so the authors will be better off calling this as relative expression.

Author Response

Reviewer #1:

The authors describe the immunomodulatory properties of iPS-derived RPE cells and its implications on interaction with immune cells from uveitis patients. The following points need attention.

  1. Line 366. 836B1, 101G26, 454E2 and TLHD1 are in fact the RPE cells derived from iPS cells, but their description on line 366 appears as if these were only the human iPS cells, and not the ones that were already converted into RPE. Related to this, it would be nice if the authors describe the details on the appearance of these cells after they become RPE like. Cuboidal morphology, distribution of gap junction proteins, or a description of similar markers will strengthen the conversion into RPE like cells.

Response: Thank you for your comments. As per your suggestions, we described the appearance of these cells, expression of gap junction proteins (ZO-1), and RPE-specific markers after they become RPE like (Revised manuscript, lines 373-376).

  1. Figure 2. The authors claim only ICAM1 and B7-H1 as the markers that were induced. They claim B7-H3 to be a weakly induced, while in fact, it is expressed at the same level as high expressers. Can the authors clarify this?

Also, the induction of ICAM1 and B7-H1 is higher in iPS derived cells, as compared with the fetal RPE cells. This deserves a mention.

Response: Thank you for your comments. We agree with your comments. So, we have changed the sentences in Fig. 2 of revised manuscript (lines 108-110).

  1. Line 121. The title of this paragraph includes only B7-H1, while the following description also has B7-DC and B7-H3 included. Why was the title restricted to only B7-H1?

Response: We agree with your comments. Therefore, we have changed the sentence in the title and the following description (Revised manuscript, lines 125-126). 

  1. The relative expression of target genes is described in the following figures (Fig 3E, Fig 5C, Fig 6C, Fig 8B, Fig S2B and Fig S2C). The authors have given Expression of (target) mRNA and in parenthesis, they have delta delta C as the title of y-axis, and used the same description in figure legends. Again, what is described here is relative expression of the specific genes, with a control arbitrarily taken as 1. In the calculation, delta delta C does not in fact give the number that is provided in the figure, so the authors will be better off calling this as relative expression.

Response: We agree with above comments. Therefore, we have changed the title of y-axis (Relative expression〜) in these figures (Fig 3E, Fig 5C, Fig 6C, Fig 8B, Fig S2B and Fig S2C).

We would like to thank the reviewer for their careful review of our manuscript and their helpful comments.

Reviewer 2 Report

In this paper authors study if RPE cells derived from human iPS could suppress or stimulate T cells from uveitis patients via co-stimulatory signals. They show that RPE cells can do both and express negative and positive co-stimulating molecules.

The study is well written and results are clearly exposed. 

Minor comment: authors are showing that the ratio TCells:RPE is an important factor in the suppressing effect of T cells by RPE. Could they be more precise in whic ratio they are using in their different experiments as it is not clearly exposed. Moreover, they could discuss what conditions could drive the RPE cells towards a suppressive or a activation mechanism (maybe the ratio of T cells or level of cytokines)

Author Response

Reviewer #2:

In this paper authors study if RPE cells derived from human iPS could suppress or stimulate T cells from uveitis patients via co-stimulatory signals. They show that RPE cells can do both and express negative and positive co-stimulating molecules.

The study is well written and results are clearly exposed. 

Minor comment: authors are showing that the ratio TCells:RPE is an important factor in the suppressing effect of T cells by RPE. Could they be more precise in which ratio they are using in their different experiments as it is not clearly exposed. Moreover, they could discuss what conditions could drive the RPE cells towards a suppressive or a activation mechanism (maybe the ratio of T cells or level of cytokines)

Response: We agree with your comments. So, we added the ratio T cells:RPE in the legends of appropriate figures. Thank you so much.